# Synthesis, Anticancer, Antioxidant, Anti-Inflammatory, Antimicrobial Activities, Molecular Docking, and DFT Studies of Sultams Derived from Saccharin

**DOI:** 10.3390/molecules27207104

**Published:** 2022-10-20

**Authors:** Nourah Al-Fayez, Hany Elsawy, Mohammed A. Mansour, Mohamad Akbar Ali, Ibrahim Elghamry

**Affiliations:** 1Department of Chemistry, College of Science Al Hufuf, King Faisal University, Al Hufuf P.O. Box 380, Saudi Arabia; 2Biochemistry Division, Department of Chemistry, Faculty of Science, Tanta University, Tanta 31527, Egypt; 3Cancer Biology and Therapy Lab, Division of Human Sciences, School of Applied Sciences, London South Bank University, London SE1 0AA, UK; 4Department of Chemistry, College of Art and Science, Khalifa University, Abu Dhabi 127788, United Arab Emirates

**Keywords:** saccharin, anti-inflammatory, antioxidant, anticancer, molecular docking, DFT calculation

## Abstract

A series of *N*-substituted saccharins namely 2-(1,1-dioxido-3-oxobenzo[d]isothiazol-2(*3H*)-yl) acetonitrile (**2**) and (alkyl 1,1-dioxido-3-oxobenzo[d]isothiazol-2(3*H*)-yl) acetate (**3a**–**g**) were synthesized, in moderate to excellent yields, from commercially available starting materials by two different approaches and their chemical structures were characterized by spectroscopic techniques (^1^H-NMR, ^13^C-NMR, IR, and MS). All the synthesized compounds were evaluated for their anti-inflammatory toward IL-6 and TNF-α, antioxidant, as well as their anticancer activities against hepatic cancer cells. In addition, their anti-fungal and antibacterial activities against both Gram-positive and Gram-negative bacteria were tested. All the tested compounds have exhibited excellent (**3a**, **d**, **e**) to moderate anti-inflammatory activity. Additionally, esters (**3b**, **f**) and nitrile (**2**) showed excellent antioxidant activity. Furthermore, ester **3f**, with isopropyl ester, exhibited the highest cytotoxic activity compared to the other esters. Moreover, all compounds were evaluated as selective inhibitors of the human COX-1 enzyme using molecular docking by calculating the free energy of binding, inhibition constant, and other parameters to find out the binding affinity. The molecular study showed that esters (**3d**, **f**) and nitrile (**2**) revealed the highest binding affinities, hence enhancing the inhibition activity with the active site of the COX-1 enzyme. All the tested compounds have more negative Gibbs free, electrostatic, and total intermolecular energies than the standard inhibitor ASA. These results indicate that, all the tested sultams are potent anti-inflammatory drugs as compared to standard inhibitors. Finally, the chemical properties and the quantum factors of synthesized sultams were calculated based on density functional theory (DFT) to predict reactivity, and then correlated with the experimental data. Ester **3f** showed the lowest ionization potential and lowest energy gap (E_gap_ = 7.5691 eV), which was correlated with its cytotoxic activity. Furthermore, the spatial electron distribution of HOMO, LUMO were computed and it clearly indicates the electron donation ability of all the tested compounds.

## 1. Introduction

Sultams are considered important compounds of heterocyclic sulfonamide due to their common uses in the industry and pharmaceuticals [1,2]. Since the first synthesis of saccharin (1,2-benziso thiazoline-3-one 1,1-dioxide) (**1**) (Figure 1) in 1879 by Remsen and Fahlbeg [3], it has been considered the most commonly used starting material for synthesis of this scaffold. Saccharin has many advantages over the other precursors. It is commercially accessible, highly soluble in water, stable, and inexpensive in mass production. Therefore, sultams derived from saccharin have been used and applied in comprehensive applications in many fields. For example, it is an intermediate in the production of pesticides and herbicides [4,5,6]. Moreover, they were proved to be a useful ingredient in the electroplating industry [7,8,9,10,11]. In addition, saccharin and its derivatives were used as catalysts and agents in organic reactions, for example *N*-bromosaccharin is employed as a catalyst, brominating agent, and oxidizing agent in various organic reactions [12]. On the other hand, saccharin and its derivatives have a broad spectrum of biological and pharmaceutical activities. They have revealed tyrosinase [13], histone deacetylase [14], kinase [15] inhibitors, antibacterial, antioxidant [16], and antianxiety [17] activities. Additionally, recently sultams have been shown to selectively bind to tumor-associated carbonic anhydrases [18].

Additionally, Ipsapirone and Repinotan are agonists of the hydroxytryptamine receptor 1A (5-HT_1A_) and are used as anti-depressant agents, derived from saccharin, either by functionalization with pyrimidinopiperazine or aminomethyl chromontively, respectively [19,20,21]. Furthermore, *N*-substituted saccharins are used as a precursor for synthesis of Piroxicam as a lead pharmaceutical to be widely used as a nonsteroidal anti-inflammatory drug (NSAIDs), Oxicams (Figure 1), to treat rheumatoid arthritis, osteoarthritis, and other acute inflammation [22,23,24]. However, these drugs have revealed some drawbacks, such as adverse skin reactions, photoallergy, and abnormal photosensitivity [25]. These drawbacks are attributed to some of the photodegradation products or metabolite end products of the drugs [25,26,27,28].

Moreover, *N*-substituted saccharins were reported to have diverse biological activities, which have been documented in many literature reports. For example, they are used for inhibition of human leukocyte elastase (HLE) [29], as antibacterial, antioxidant [30], and antipyretic agents [31], and for inhibition of hcA I, II, IX, and XII [32,33] (Figure 2).

Despite the numerous studies which can be found in the literature dealing with synthesis and examination of biological activities of this scaffold, as mentioned above, thorough studies and inspection are still required in both chemistry and biological revaluation of old and new sultams derived from the commercially available saccharin. Therefore, and in continuation of our comprehensive and deep study of the chemistry and biological evaluation of sultams derived from saccharin [16,34,35,36], herein we report the synthesis of a series of *N*-substituted saccharins (**2**, **3a**–**g**) and study their anti-inflammatory, antioxidant, cytotoxic, antibacterial, and fungal activities. In addition, all compounds are evaluated as selective inhibitors of the COX-1 enzyme by using molecular docking calculations. Finally, a quantum chemical calculation study is carried out to provide insight on the chemical properties of the synthesized compounds, then correlated with the experimental data.

## 2. Results and Discussion

The most common method of synthesis of the *N*-alkyl saccharins in literature is the direct alkylation of either sodium or potassium salt of saccharin with alkyl halide in DMF. Therefore, different alkyl side chains, with either electron-withdrawing or electron-donating groups, can easily be bonded to the nitrogen atom in the isothiazaole ring of the saccharin moiety. Hence, a variety of *N*-alkyl saccharins with nitrile, ester alkoxy, and heteroaryls have been prepared by this method in refluxing dimethylformamide [37,38,39,40,41,42]. In some literature reports, sodium or potassium salts of saccharin can be prepared in situ by heating the saccharin with the alkyl halides in DMF directly [32]. Furthermore, the *N*-alkyl saccharin derivatives have been obtained in the presence of sodium hydride in dimethylformamide or ionic liquid of butyl pyridinium terafluoroborate under microwave conditions [43,44,45].

In this work, the 2-(1,1-dioxido-3-oxobenzo[d]isothiazol-2(*3H*)-yl) acetonitrile **2** (Figure 1) was synthesized following modified literature conditions by heating the saccharin sodium salt **1** with bromoactanitrile in DMF at 80–90 °C. The reaction was completed with excellent yield (96%) of high purity compared to the previously reported method (66%, 72%, 82%, 88%) [32,37,38]. Whereas, 2-(1,1-dioxido-3-oxobenzo[d]isothiazol-2(*3H*)-yl) acetate derivatives **3a**–**g** were obtained in moderate to excellent yields following a modified literature procedure [38] by acidic hydrolysis of 2-(1,1-dioxido-3-oxobenzo[d]isothiazol-2(*3H*)-yl) acetonitrile **2** with concentrated sulfuric acid and excess different alcohols (methanol, ethanol butanol, pentanol, propan-2-ol, pentan-2-ol, respectively) at 100 °C (Figure 1). Furthermore, esters **3d**–**g** were synthesized by an alternative method of refluxing of **3b** in concentrated hydrochloric acid at 120 °C, followed by hydrolysis of **4** with different alcohols in concentrated sulfuric acid at 120 °C (Figure 2).

The chemical structures of **2** and **3a**–**g** were determined by spectroscopy (^1^H-NMR, ^13^C-NMR, IR, and MS). Thus, the ^1^H-NMR spectra of compounds **2** and **3a**–**g** revealed singlet peak in the range of 4.37–4.65 ppm, which is attributed to the two protons of the side chain methylene group. A broad singlet for protons at 0.88–4.30 and 5.97–6.06 ppm were assigned to the ester groups observed in Table 1. The^13^ C-NMR spectra revealed two down-field signals at δ = 157.83–158.84 ppm which are attributed to the carbonyl groups of amidic groups, and δ = 165.89–166.43 ppm which were assigned to the carbonyl carbons of ester groups **3a**–**g**, whereas δ = 112.79 ppm which was assigned to carbon in cyanide group of **2**. In addition, the IR spectra of **3a**–**e** revealed two singlets at ν = 1730 to1744 cm^−1^ and 1772 to 1750 cm^−1^ which were attributed to amidic and ester carbonyl groups. Similarly, the spectra of **3f**–**g** revealed two singlets at ν = 1730 and 1646–1663 cm^−1^ which were attributed to amidic and ester carbonyl groups, while **2** revealed a singlet peak at ν = 1741 cm^−1^ which was assigned to amidic carbonyl and ν = 2021 cm^−1^ which was attributed to the cyanide group. Furthermore, mass spectroscopic data confirmed the molecular formula of the synthesized sultams (**2**, **3a**–**g**). In addition, mass spectra of the nitrile **2** revealed a characteristic fragment at *m*/*z* = 158 accounting for the loss of sulfur dioxide at the thiazole ring. Ester **3a**–**3b** and **3e**–**3g** revealed a fragment at *m*/*z* = 196 accounting for the loss of the ester group, whereas ester **3c**–**3d** showed a fragment at *m*/*z* = 237 accounting for the loss of the alkyl side chain at the ester group.

The synthesized sultams **2** and **3a**–**g** were selected to evaluate their in vivo anti-inflammatory and their in vitro antioxidant, anticancer, antibacterial, and antifungal activities. Moreover, their inhibitors of COX-1 have been evaluated.

### 2.1. Anti-Inflammatory Studies

Inflammation is a significant mechanism to defend the body against infection or any physical or chemical functions, and an inflammatory environment is generated as the result of an imbalance between pro-inflammatory and anti-inflammatory systems [46]. Among the significant mediators of inflammation in the liver, many pro-inflammatory cytokines such as interleukin-6 (IL-6), and tumor necrosis factor-alpha (TNF-α) are released from macrophages. Activation of TNF-α and IL-6 are correlated with a different disease condition, and a cytotoxic protein is secreted by macrophages in response to inflammation [46]. Consequently, all the synthesized compounds **2** and **3a**–**g** were screened for anti-inflammatory activity, by measuring IL-6 and TNF-α in the presence of CCl_4_-induced liver damage and has been used to evaluate their therapeutic potential. In addition, this model has been applied to mechanical liver injuries that are comparable to human liver disease in terms of both the morphology and the biochemical characteristics of the lesions [47]. The results of the tested compounds are given in Table 2. It has been found that all compounds under investigation exhibited the anti-inflammatory activates toward both IL-6 and TNF-α. Esters **3a**, **3d**, and **3e** with either the shortest alkyl chain at the ester side (Me), or with the longer chain, butyl and n-pentyl respectively, exhibited the highest anti-inflammatory activities toward both IL-6 and TNF-α. Hence, the length of the alkyl side chain has no effect on the anti-inflammation effect of such systems. However, the other esters with branched alkyl side chain esters, **3f** and **3g,** showed the lowest activity compared with either the provirus esters (**3a**, **3d**, and **3e**) or saccharin bonded to the alkyl side chain with the nitrile group (**2**).

### 2.2. Antioxidant and Anticancer Studies

Free radicals and reactive oxygen species (ROS) that include hydroxyl, superoxide anions, and hydrogen peroxide are natural products in oxidative reaction in metabolism [48,49,50]. At low cellular concentrations, ROS have beneficial effects in physiological roles such as in cellular response, for example in defense against infectious agents [51]. In contrast, high cellular concentrations have harmful effects. They are generated in response to cytokine-induced stress signals [46]. The generated free radicals can react with a wide range of biological substrates, e.g., lipids, DNA, and proteins [51] creating an oxidative stress in normal cells which are closely related to numerous health disorders such as cancer [52,53,54].

Hydrogen peroxide, which is rapidly decomposed into oxygen and water, leads to hydroxyl radicals (OH^.^) which can produce lipid peroxidation and cause DNA damage [55]. The result of hydrogen peroxide scavenging activity of the synthesized compounds (**2** and **3a**–**g**) is presented in Table 3. It was found that all the compounds under examination exhibited antioxidant activities. Ester **3b** (Et) and compound **2** exhibited the highest antioxidant activities with values of 41 and 39, respectively, compared to the reference material. Esters with a branched side chain (**3f**, **g**) exhibited moderate antioxidant activity compared to the esters with a normal side chain structure (**3c**–**e**).

Moreover, the MTT experiment was used to study the cytotoxic activity of the synthesized compounds by using hepatic cancer cells and the results are given in Table 4. The result showed that compound **2** and ester **3f**, with an isopropyl side chain in the ester end, exhibited the highest cytotoxic activity with inhibition rates of 60% and 55%, respectively. While esters **3b**, **d**, **a** exhibited the lowest cytotoxic activity with inhibition rates of 5%, 7.5%, and 12.5%, respectively. The other esters (**3c**, **g**) showed moderate activities with inhibition rates of 47.5% and 42.5%, respectively.

### 2.3. Antimicrobial Activities

The assessment of antibacterial activity of **2** and **3a**–**g** were determined by testing against Gram-negative *Pseudomoas aeruginosa* (*P. aeruginosa*), *Escherichia coli* (*E. coli*) and *Klebsiella pneumoniae* (*K. peneumoniae*), and Gram-positive *staphylococcus aureus* (*S. aureus*) and *streptococcus pneumonia* (*Streptococcus* spp), also one fungal strain *Candida Albicans* (*C. Albicans*). As the compounds under investigation are considered as derivatives of sulfonamide, one of the sulfa drug antibiotics, namely Sulfamethoxazole (SMZ or SMX) was used as a positive control. The test showed that, all the compounds under investigation did not exhibit any antifungal or antibacterial activities toward both the Gram-negative and Gram-positive strains.

### 2.4. Molecular Docking

Compounds **2** and **3a**–**g** were tested as anti-inflammatory ligands (Table 2) where the pro-inflammatory cytokines IL-6 and TNF-α were investigated. Therefore, we tested the inhibitory effect of these drugs against inflammatory modulators such as cyclooxygenases (COXs). COXs enzymes are responsible for conversion of arachidonic acid into the prostaglandin E2 (PGE2), which is the main moderator of the inflammation process [56]. In order to compare the inhibitory effect of these drugs, we used acetylsalicylic acid (ASA) as the standard COX-1 inhibitor. In addition, the free energy of binding, inhibition constant (Ki), total energy of vdW + Hbond + desolv (E_VHD_), electrostatic energy, total intermolecular energy, frequency of binding, and interact surface were evaluated to estimate the favorable binding of ligand molecules to enzymes.

The docking results of the synthesized compounds **2**, and **3a**–**g** is presented in Table 5. Additionally, decomposed interaction energies with hydrogen bonds, polar, hydrophobic, and other bonds are given in Table 6.

The study showed that all the values of the free binding energy of all compounds (**2**, and **3a**–**g**) are negative, in the range −3.26 to −3.89, which indicates that the interaction process is commonly exothermic. All tested compounds (compounds **2** and **3a**–**g**) have more negative Gibbs free, electrostatic, and total intermolecular energies than the standard inhibitor ASA. Additionally, all the synthesized compounds (**2**, and **3a**–**g**) were found to be inhibitory by occupying some of the active sites in the COX-1 with hydrogen, polar, or other bonds. Moreover, the value of the inhibition constants indicates the efficiency of compounds to inhibit the enzyme, and low free energy with low Ki values proves drug efficiency [57]. Esters **3d**, **3f** and compound **2** exhibited low Ki values and low energy binding values. They are considering the highest inhibition activity of COX-1 among all compounds as displayed in Table 5. Ester **3d**, with the normal butyl ester exhibit the highest inhibition activity, with binding energy value of −3.89 kcal/mol and inhibition constant of 1.40 µM. Moreover, it has the lowest E_VDW,_ and total energy interaction values of −5.45 kcal/mol and −5.47 kcal/mol. The high inhibition efficiency appeared with the interaction surface of over 400 is displayed in Table 5. It was found that Tyr55 and Asn68 formed polar bonds with ester **3d**. In addition, hydrophobic bond interactions were found with Tyr38, Pro40, and Pro35 are displayed in Table 6. The 3D/2D interacting docking of ester **3d**, which are selected as the best interaction esters with COX-1 is displayed in Figure 3, whilst the other compound is displayed in Appendix A. An HB plot of compound **3d** was generated to show the interactions with the enzyme and displayed in Figure 4, the other compounds are displayed in Appendix A.

On the other hand, esters **3c**, **3e**, and **3g** exhibited low inhibition compared to other compounds (**2**, **3d**, and **3f**). Although they showed hydrogen bond interactions with Tyr55 [58] in COX-1, which is considered the strongest interaction bond. The hydrogen bond was identified between the oxygen atoms of sulfonamide and oxygen of Tyr55 in COX-1 (2.88, 3.34, and 3.31 Å, respectively,). In addition, hydrophobic bond interactions were found with Tyr38, and Pro40 (for compounds **3c**, **g**), with Tyr38, Pro40, and Pro35 (for compound **3e**). Other bond interactions were found also with Pro35 (for compound **3c**, **g**), and with Asn68 (for compound **3e**) as displayed in Table 6. Compound **3g**, with an isopentyl ester exhibited the lowest inhibition activity of COX-1, with a binding energy of −3.26 kcal/mol and inhibition constant of 4.09 µM. Moreover, the E_VDW_ and total energy interaction values are −4.22 kcal/mol and −4.23 kcal/mol as displayed in Table 5. The 3D/2D interacting docking of Compound **3g**, which was selected as the lowest interaction with COX-1 is displayed in Figure 4, whereas the other compounds are displayed in Appendix A. An HB plot of compounds **3g** was given in Figure 5, and other compounds are displayed in Appendix A. Esters **3c**, and **3g** have comparable decomposed interactions with the active site of COX-1, but they have different values of bond free energy and inhibition constants (Ki).

Furthermore, esters **3a** and **3b**, containing methyl, and ethyl groups respectively, exhibited moderate inhibition activity with binding energy values of −3.58 and −3.49 kcal/mol and inhibition constants of 2.39 and 2.77 µM. The E_VDW_ values are −4.42 and −4.46, and the total energy interaction values −4.40 and −4.45 kcal/mol as in Table 5. Both esters were found to interact with Tyr55 and Tyr38. The hydrophobic bonds were formed with Pro35, and Tyr38 for ester **3a** whereas, ester **3b** interacted with Pro35, and Pro40 as displayed in Table 6. The 3D/2D interacting docking of ester **3a**, with COX-1 is displayed in Figure 4, and an HB plot is displayed in Figure 5, while 3D/2D and HB plot of ester **3b** is displayed in Appendix A.

### 2.5. Computational Details

The optimized structures of the synthesized compounds **2** and **3a**–**g** were calculated using M062X/6-31 + G(d,p) level and the optimized structure of compounds are represented in Figure 5.

The ***HOMO*** (highest occupied molecular orbital) and ***LUMO*** (lowest unoccupied molecular orbital) energy values were used to determine the charge transfer within the molecule and the energy gap (***E_HOMO_*** − ***E_LUMO_***) gives information about the chemical reactivity and kinetic stability of the molecule. The ***HOMO*** orbital acts as an electron donor, while the ***LUMO*** orbital acts as the electron acceptor. These molecular orbitals are also called the frontier molecular orbitals (FMOs). From the analysis of ***HOMO*** and ***LUMO*** energy, the other chemical properties such as ionization potentials (***I***), electron affinities (***A***), chemical softnesses (***S***), chemical hardnesses (***η***)**,** electronic chemical potentials (**µ**), and electronegativities (***χ***) were calculated by the following Equations 1–6 [59,60,61], and the calculated values for the compounds are given in Table 7.
(1)$$l=-E_{HOMO}$$
(2)$$A=-E_{LUMO}$$
(3)$$\chi =\left(l+A\right)/2$$
(4)$$\eta =\left(l-A\right)/2$$
(5)S=1/η
(6)$$µ=-\left(l+A\right)/2$$

Small ***E_HOMO_*** − ***E_LUMO_*** gaps correspond to more chemical stability and the charge transfer easily occurs in it which enhances its biological activities [62]. In this study, energy gaps of the synthesized compounds (**2**, **3a**–**g**) are calculated in the following order: **2** > **3a** > **3c** > **3b** > **3d** > **3e** > **3g** > **3f**. Consequently, ester **3f** which has the lowest energy gap (**E_gap_** = 7.5691 eV) is the most stable compound, while nitrile **2** has the highest energy gap (**E_gap_** = 7.9874 eV) andis the less stable one. The ionization potential (I) of a compound is directly affected by the ***HOMO*** energy. The higher ***HOMO*** energies correspond to higher electron-donating. In this study, the lowest ionization potential and the highest ***HOMO*** energy were calculated for ester **3f**, with the isopropyl side chain in the ester end, (***I*** = 9.0525 eV) (***E_HOMO_***= −9.0525 eV), meaning that the electron-donating effect is the highest compared to the other esters. In contrast, electron affinity (***A***) of compounds is directly affected by the ***LUMO*** energy, and the lower ***LUMO*** energies correspond to higher electron-accepting. In this study, the highest electron affinity (***A***) and the lowest LUMO energy were calculated for compound **2** (**A** = 1.7612 eV) (***E_LUM_****_O_* = −1.7612 eV), hence, it is a good electron acceptor.

Moreover, the sensitiveness of the system’s energy to charge in the number of electrons is described by electronic chemical potential (**µ**). In this study, a more negative electronic chemical potential value was obtained for compound **2** (**µ** = −5.7549 eV). Chemical hardness is used to measure the resistance to charge transfer. In this study, the chemical hardness of the synthesized compounds (**2**, **3a**–**g**) is calculated in the following order: **2** > **3a** > **3c** > **3b** > **3d** > **3e** > **3g** > **3f**, and chemical softness is the inverse of hardness. Ester **3f**, with the isopropyl side chain in the ester end exhibited the lowest hardness (***η*** = 3.7845 eV) and the highest softness (***S*** = 0.2642 eV^−1^), the smallest energy gap, and the highest softness with more polarity. Consequently, it will be more reactive than all the other esters, because it needs small excitation energies which influence the biological activity of the molecule [62]. It was noticed throughout this study, ester **3f** showed the highest anti-cancer activity compared to the other esters. While it exhibited moderate and low antioxidant and anti-inflammatory activities, respectively, compared to the other esters. Furthermore, the spatial electron distribution of ***HOMO***, ***LUMO*** were computed and it clearly indicates the electron donation ability. The spatial electron distribution of each compound (**2**, **3a**–**g**) are shown in Figure 6.

## 3. Materials and Methods

All reagents and solvents used in this work were purchased from commercial sources without further purification. Melting points were determined with a Gallenkamp and uncorrected. IR spectra were recorded on Bruker-Alpha spectrometer. Absorption maxima ν_max_ are reported in wavenumber cm^−1^. ^1^H NMR and ^13^C NMR spectra were recorded on Bruker (400 MHz) using CDCl_3_ as the solvent. Chemical shifts are defined as δ values in ppm (parts per million) referenced to the residual solvent signal (^1^H 7.26; ^13^C 77.2). ^1^H-NMR spectra are described as follows: δ chemical shift/ppm (multiplicity, J-coupling, number of protons). ^13^C-NMR spectra are described as follows: δ chemical shift/ppm (assignment). Multiplicity is defined as follows: s, singlet; d, doublet; t, triplet; q, quartet; m, multiple; dd, doublet of doublets. Coupling J is valued in hertz (Hz). Mass spectra were determined on a Varian AMD 604 instrument using 70 eV ionization energy with methanol as the solvent. Progress of the reactions were monitored by thin-layer chromatography (TLC) with UV light (254–365 nm).

### 3.1. Synthesis of 2-(1,1-Dioxido-3-oxobenzo[d]isothiazol-2(3H)-yl) acetonitrile (***2***)

A mixture of saccharin sodium salt (**1**) (0.02 mol) and bromoactanitrile (0.02 mol) in 10 mL DMF was heated at 80–90 °C in an oil bath for 2 h. The reaction mixture was cooled to room temperature, then poured into ice-water. The resulting solid product was filtered, washed with water several times, air-dried, and crystallized from ethanol/water.

White crystals; Yield: 96.30%; mp:146–146.3 °C (Lit.mp: 130–134 °C [32] /142–143 °C [37]); IR ν_max_:3090(ν C_sp2_-H), 2991(ν C_sp3_-H), 2021(ν CN), 1741(ν C=O), 1336(ν_as_ S=O), 1256(ν C-N), 1182(ν_s_ S=O) cm^−1^. ^1^H-NMR (CDCl_3_, 400 MHz): δ 4.65 (s, 2H), 7.92–8.15 (m, 4H, Ar-H); ^13^C NMR (CDCl_3_, 400 MHz): δ 25.19 (CH_2_), 112.79 (CN), 121.53 (Ar), 125.89 (Ar), 126.39 (Ar), 135.00 (Ar), 135.80 (Ar), 137.53 (Ar), 157.83 (C-N); MS: *m*/*z* (%) = 222 (M^+^,20), 158 (60), 142 (20), 130 (22), 105 (22), 76 (70). Calcd. for C_9_H_6_N_2_O_3_S (222.22): C, 48.65; H, 2.72; N, 12.61; O, 21.60; S, 14.43.

### 3.2. General Procedure for Synthesis of 2-(1,1-Dioxido-3-oxobenzo[d]isothiazol-2(3H)-yl) acetate (***3a***–***g***)

The acetonitrile compound (**2**) (0.01 mol) was added to solution of conc_._ sulfuric acid (0.02 mol) in the different alcohols (2.7 mL) at 0 °C. After the addition was completed, the mixture was heated under reflux at 100 °C in oil bath for 6–7 h. The reaction was cooled at room temperature, then diluted with ethyl acetate (60 mL). The organic layer was washed with water (27 mL), sodium bicarbonate solution, and brine (27 mL), and dried over anhydrous sodium carbonate. The solvent was removed under vacuum and the solid obtained was crystallized from either methanol or ethanol. The physical constants and the spectral data of sultams **3a**–**g** are given below.

#### 3.2.1. Methyl 2-(1,1-dioxido-3-oxobenzo[d]isothiazol-2(3H)-yl) acetate (**3a**)

White crystals from methanol; Yield: 68.93%; mp: 123–124 °C (Lit.mp: 113−115 °C [40], 116–117 °C [41,44]); IR ν_max_: 3085(ν C_sp2_-H), 2963(ν C_sp3_-H), 1772(ν C=O), 1744(ν C=O), 1336(ν _as_ S=O), 1267(ν C-N), 1217(ν C-O), 1185(ν_s_ S=O) cm^−1^. ^1^H NMR (CDCl_3_, 400 MHz): δ 3.81 (s, 3H), 4.47 (s, 2H), 7.84–8.10 (m, 4H, Ar-H); ^13^C NMR (CDCl_3_, 400 MHz): δ 38.92 (CH_2_), 52.99 (CH_3_), 121.25 (Ar), 125.48 (Ar), 125.03 (Ar), 134.57 (Ar), 135.17 (Ar), 137.72 (Ar), 158.77 (C-N), 166.44 (C-O); MS: *m*/*z* (%) = 225 (M^+^,5), 196 (100), 169 (10), 132 (10), 104 (20), 77 (30). Calcd. for C_10_H_9_NO_5_S (255.24): C, 47.06; H, 3.55; N, 5.49; O, 31.34; S, 12.56%.

#### 3.2.2. Ethyl 2-(1,1-dioxido-3-oxobenzo[d]isothiazol-2(3H)-yl) acetate (**3b**)

White crystals from ethanol; Yield: 78%; mp:106–107.6 °C (Lit.mp: 105–106 °C [44]); IR ν_max_: 3093(ν C_sp2_-H), 2963(ν C_sp3_-H), 1750(ν C=O), 1730(ν C=O), 1336(ν_as_ S=O), 1267(νC-N), 1217(ν C-O), 1183(ν_s_ S=O) cm^−1^. ^1^H NMR (CDCl_3_, 400 MHz): δ 1.32 (t, 3H), 4.29 (q, 2H), 4.48 (s, 2H), 7.85–8.13 (m, 4H, Ar-H); ^13^C NMR (CDCl_3_, 400 MHz): δ 14.07 (CH_3_), 39.14 (CH_2_), 62.30 (CH_2_), 121.24 (Ar), 125.49 (Ar), 127.11 (Ar), 134.52 (Ar), 135.10 (Ar), 137.79 (Ar), 158.81 (C-N), 165.90 (C-O); MS: *m*/*z* (%) = 269 (M^+^,5), 224 (5), 196 (100), 169 (20), 133 (11), 105 (50), 76 (30). Calcd. for C_11_H_11_NO_5_S (269.27): C, 49.07; H, 4.12; N, 5.20; O, 29.71; S, 11.91%.

#### 3.2.3. Propyl 2-(1,1-dioxido-3-oxobenzo[d]isothiazol-2(3H)-yl) acetate (**3c**)

White crystals from ethanol; Yield: 71%; mp: 105–106 °C; IR ν_max_: 3085(νC_sp2_-H), 2969(νC_sp3_-H), 1750(ν C=O), 1730(ν C=O), 1336(ν_as_ S=O), 1293(ν C-N), 1211(ν C-O), 1183(ν_s_ S=O); ^1^H NMR (CDCl_3_, 400 MHz): δ 0.94 (t, 3H), 1.69 (m, 2H), 4.16 (t, 2H), 4.47 (s, 2H), 7.84–8.10 (m, 4H, Ar-H); ^13^C NMR (CDCl_3_, 400 MHz): δ 10.23 (CH_3_), 21.67 (CH_2_), 39.10 (CH_2_), 67.77 (CH_2_), 121.23 (Ar), 125.45 (Ar), 127.05 (Ar), 134.55 (Ar), 135.14 (Ar), 137.75 (Ar), 158.81 (C-N), 166.00 (C-O); MS: *m*/*z* (%) = 282 (M^+^-H,25), 267 (1), 253 (1), 237 (13), 133 (2), 105 (2), 73 (12). Calcd. for C_12_H_13_NO_5_S (283.30): C, 50.88; H, 4.63; N, 4.94; O, 28.24; S, 11.32%.

#### 3.2.4. Butyl 2-(1,1-dioxido-3-oxobenzo[d]isothiazol-2(3H)-yl) acetate (**3d**)

White crystals from ethanol; Yield: 81%; mp: 77.1–77.8 °C; IR ν_max_: 3094(ν C_sp2_-H), 2961(ν C_sp3_-H), 1758(ν C=O), 1730(ν C=O), 1332(ν_as_ S=O), 1217(ν C-N), 1209(ν C-O), 1183(ν_s_ S=O) cm^−1^; ^1^H NMR (CDCl_3_, 400 MHz): δ 0.93 (t, 3H), 1.39 (m, 2H), 1.66 (m, 2H), 4.25 (t, 2H), 4.46 (s, 2H) 7.86–8.12 (m, 4H, Ar-H); ^13^C NMR (CDCl_3_, 400 MHz): δ 13.61 (CH_3_), 18.95 (CH_2_), 30.44 (CH_2_), 39.13 (CH_2_), 66.11 (CH_2_), 121.24 (Ar), 125.47 (Ar), 127.11 (Ar), 134.51 (Ar), 135.10 (Ar), 137.80 (Ar), 158.81 (C-N), 165.98 (C-O); MS: *m*/*z* (%) = 297 (M^+^,6), 281 (9), 267 (6), 251 (1), 237 (2), 163 (16), 133 (2), 73 (13). Clacd. for C_13_H_15_NO_5_S (297.33): C, 52.52; H, 5.09; N, 4.71; O, 26.90; S, 10.78%.

#### 3.2.5. Pentyl 2-(1,1-dioxido-3-oxobenzo[d]isothiazol-2(3H)-yl) acetate (**3e**)

White crystals from ethanol; Yield: 67.70%; mp: 82.6–83 °C; IR ν_max_: 3095(ν C_sp2_-H), 2961(ν C_sp3_-H), 1756(ν C=O), 1730(ν C=O, 1330(ν_as_ S=O), 1271(ν C-N) 1219(ν C-O), 1183(ν_s_ S=O) cm^−1^; ^1^HNMR (CDCl_3_, 400 MHz): δ 0.90 (s, 3H), 1.34 (s, 2H), 1.67 (m, 4H), 4.22 (t, 2H), 4.46 (s, 2H) 7.85–8.11 (m, 4H, Ar-H); ^13^C NMR (CDCl_3_, 400 MHz): δ 13.90 (CH_3_), 22.20 (CH_2_), 27.84 (CH_2_), 28.09 (CH_2_), 66.38 (CH_2_), 39.13 (CH_2_), 121.23 (Ar), 125.45 (Ar), 127.09 (Ar), 134.52 (Ar), 135.11 (Ar), 137.79 (Ar), 158.81 (C-N), 165.97 (C-O); MS: *m*/*z* (%) = 312(M^+^+1,5), 300 (5), 224 (20), 196 (100), 169 (10), 133 (55), 104 (30), 77 (20). Calcd. for C_14_H_17_NO_5_S (311.35): C, 54.01; H, 5.50; N, 4.50; O, 25.69; S, 10.30%.

#### 3.2.6. Isopropyl 2-(1,1-dioxido-3-oxobenzo[d]isothiazol-2(3H)-yl) acetate (**3f**)

White crystals from ethanol; Yield: 16.97%; mp:162–163 °C (lit. mp: 118–119 °C [44]); IR ν_max_: 3087(ν C_sp2_-H), 2979(ν Csp^3^-H), 1730(ν C=O), 1663(ν C=O), 1332(νas S=O), 1304(ν C-N), 1251(ν C-O), 1183(ν_s_ S=O) cm^−1^; ^1^H NMR (CDCl_3_, 400 MHz): δ 1.16 (d, 6H,2CH_3_), 4.37 (s, 2H), 6.06 (s, 1H), 7.89–8.13 (m, 4H, Ar-H); ^13^C NMR (CDCl_3_, 400 MHz): δ 22.47 (CH_3_), 41.59 (CH_2_), 42.10 (CH), 121.35 (Ar), 125.71 (Ar), 126.94 (Ar), 134.79 (Ar), 135.37 (Ar), 137.57 (Ar), 158.89 (C-N), 164.27 (C-O); MS: *m*/*z* (%) = 283(M, 89), 255 (5), 224 (20), 196 (70), 133 (30), 104 (65), 76 (100). Calcd. for C_12_H_13_NO_5_S (283.30): C, 50.88; H, 4.63; N, 4.94; O, 28.24; S, 11.32%.

#### 3.2.7. Isopentyl 2-(1,1-dioxido-3-oxobenzo[d]isothiazol-2(3H)-yl) acetate (**3g**)

White crystals from ethanol; Yield: 97%; mp:180–181 °C; IR ν_max_: 3087(ν C_sp2_-H), 2957(ν C_sp3_-H), 1730(ν C=O), 1646(ν C=O), 1342(ν_as_ S=O), 1285(ν C-N), 1257(ν C-O)cm^−1^; ^1^H NMR (CDCl_3_, 400 MHz): δ 0.88 (t, 3H), 1.13 (d, 3H), 1.39 (m, 2H), 1.54 (q, 2H), 4.42 (d, 2H), 5.96 (dd, J = 8.3,J = 9.0 Hz, 1H), 7.89–8.14 (m, 4H, Ar-H); ^13^C NMR (CDCl_3_, 400 MHz): δ 10.06 (CH_3_), 19.05 (CH_2_), 20.65 (CH_2_), 38.78 (CH_2_), 45.70 (CH_2_), 52.68 (CH_2_) 121.33 (Ar), 125.68 (Ar), 126.93 (Ar), 134.78 (Ar), 135.37 (Ar), 137.59 (Ar), 158.87 (C-N), 164.90 (C-O); MS: *m*/*z* (%) = 310 (M^+^-H,2), 295 (2), 281 (80), 224 (30), 196 (100), 133 (35), 104 (25), 77 (22). Calcd. for C_14_H_17_NO_5_S (311.35): C, 54.01; H, 5.50; N, 4.50; O, 25.69; S, 10.30%.

### 3.3. Synthesis of 2-(1,1-Dioxido-3-oxobenzo[d]isothiazol-2(3H)-yl) acetate (***3c***–***g***) from ***3b***

A mixture of the ester compound (**3b**) (7 × 10^−4^ mol) with hydrochloric acid (10 mL) was heated under reflux in oil bath for 3 h at 120 °C. The reaction mixture was poured into ice-water. The solid was collected by filtration to give crude carboxylic acid (**4**), which was used directly in the next step.

A mixture of the carboxylic acid (**4**) (2.76 × 10^−5^ mol), the different alcohols (1 mL), and 3 drops of sulfuric acid was heated under reflux in oil bath for 5 h at 120 °C. The reaction mixture was poured into ice-water. The resulting solid product was filtered, washed with water, and air-dried to give crude esters (**3c**–**g**).

### 3.4. In Vivo Assay

#### 3.4.1. Materials

CCl_4_ (Cat. No. 56-23-5) was purchased from Loba Chemie (India), Human TNFα ELISA Kit (Cat. No. EA100365) was purchased from OriGene Technologies Inc., Rockville, MD, USA, Rat Interleukin 6 (IL-6) ELSA kit (Cat. No. MBS726707) was obtained from MyBioSource, Inc., San Diego, CA, USA; total antioxidant capacity kit (Cat. No. TA2512) was purchased from Biodiagnostic, Giza, Egypt.

#### 3.4.2. Animals

Wistar male rats weighing 200–220 g were used. All experimental animals were provided from the faculty of science at King Faisal University, Kingdom of Saudi Arabia with ethical standards (Ref. No. KFU-REC/2020-09-02). All animals were housed in the laboratory animal room under standard management conditions (temperature 20–25 °C with exposure time to light per day was 12 h). Experiments were carried out using 10 animals per group.

#### 3.4.3. Anti-Inflammatory and Antioxidant Evaluation

The 20 Wistar rats were divided into two main groups. The first group: 2 rats were used as a control group and were injected i.p with olive oil at dose of 3 mL/kg body weight while the second group: 18 rats were injected i.p with CCl_4_ 30% in olive oil at dose of 3 mL/kg body weight [63]. Animals were injected i.p twice per week for seven weeks and hepatotoxicity animal models were generated. Then, the second group was subdivided into two subgroups into 9 subgroups. One group considered as CCL_4_ reference group and each group was injected i.p with 50 µM tested compounds, which were dissolved in DMSO in 1% final concentration in saline. Blood samples were collected from animals, followed by serum being separated from blood samples to test for anti-inflammatory and antioxidant activities.

Serum interleukin IL-6 (Catalog No. MBS726707, MyBioSource, Inc., San Diego, CA, USA) and tumor necrosis factor TNF-α (Catalog No. EA100365, OriGene Technologies Inc., Rockville, MD, USA) levels were measured (Table 2) to determine the anti-inflammatory activity of 2 and 3a–g by using kit according to the manufacturer’s instruction. Total antioxidant capacity was assayed following the protocol accompanied with kit (Catalog no. TA2512, Biodiagnostic, Giza, Egypt). Briefly, the estimation of the antioxidant capacity was carried out by the reaction of compounds with a defined amount of exogenously provided hydrogen peroxide (H_2_O_2_). The residual H_2_O_2_ was estimated calorimetry. Total antioxidant concentration was expressed as mM/L (Table 3)

### 3.5. In Vitro Assay

#### 3.5.1. MTT Assay

The anticancer activity of **2** and **3a**–**g** was investigated against hepatic cancer cells (Hepatocellular carcinoma (HEPG-2) cells were obtained from ATCC via Holding company for biological products and vaccines (VACSERA), Cairo, Egypt) with MTT (3-(4,5-Dimethylthiazol-2-yl)-2,5 diphenyltetrazolium bromide) assay completed according to a method developed by Mitry et al. [64]. The cell lines were seeded in a 96-well plate at a density of 1.0 × 104 cells/well at 37 °C for 48 h under 5% Co2. After incubation, the cells were treated with different concentrations of compounds and incubated for 24 h. MTT was added, followed by an incubation for 4 h at 37 °C. In this method, yellow MTT is reduced formazan in the mitochondria of viable cells. The insoluble purple formazan product is dissolved in acidified isopropanol, then anti-cancer activity was measured using microplate reader (Reader Xl800, DIALAB, Neudorf, Austria) at 570 nm. The inhibition rate of cell growth (*IRCG*) was calculated by the following equation:(7)$$IRCG=\frac{Mean\;value\;of\;treated\;group}{Conttrol\;group}\times 100$$

#### 3.5.2. Antibacterial and Antifungal Evaluation

The antibacterial and antifungal activities of test compounds were evaluated using agar diffusion method. The antibacterial screening was carried out against typical bacterial strains of Gram-negative *P. aeruginosa*, *E. Coli*., and *K. pneumoniae*. and Gram-positive *S. aureus*. and *Streptococcus* ssp. In addition, one fungal strain *C. Albicans*. The antimicrobial investigation on the test compounds was performed as previously described [65]. During the antibacterial study, Sulfamethoxazole (SMZ or SMX) was used as a positive control for the bacterial strains, whereas Nystatin was selected as a fungal reference.

#### 3.5.3. Docking

A docking study of all the synthesized compounds **2**, and **3a**–**g** were performed on COX-1 enzyme (Protein Data Bank (PDB): 3n8w—OXIDOREDUCTASE). Docking calculations were carried out using Dockingserver [66]. The Merch molecular force field 94 (MMFF 94) was used for energy minimization of ligand molecular **2** and **3a**–**g** using Dockingserver [67]. Non-polar hydrogen atoms were merged, and Gasteiger partical charges were added to ligand. Furthermore, rotatable bonds were defined in ligand bonds. Docking calculations were carried out on 3n-OXIDOREDUCTASE protein model. Essential hydrogen atom, Kollman united atom type were added using AutoDouk tools [67]. AutoGrid program was then used to generate grid maps by neutralizing the protein, a 20 × 20 × 20 dimensions with 0.375 Å distance was used. AutoDock parameter set-and distance-dependent dielectric function was used in the analysis of docking results, determination of the Van der waals and electrostatic terms, respectively. The Lamarckian genetic algorithm (LGA), and the Solis & Wets local search method [68] were used to perform docking simulations. Random initial position, orientation, and torsions of ligand were used during the procedure. Each docking experiment was derived from 2 different runs that were set to terminate after a maximum of 250,000 energy evaluations. The population size was set to 150. During the search, a translational step of 0.2 Å, and quaternion and torsion steps of 5 were applied.

#### 3.5.4. Computational Details

The quantum chemical calculations of synthesized compounds **2** and **3a**–**g** were carried out by using the Gaussian 09 software (Gaussian, Inc., Wallingford, CT, USA) [69]. All the reactants, intermediates, and products were optimized using the hybrid DFT method with Minnesota functional of M06-2X and 6-31 + G (d,p) as the basis set [69]. The vibrational frequencies were calculated on optimized geometries. All the reactants, intermediates, and products have zero imaginary frequency, which confirms that structures are optimized on their global minimum.

## 4. Conclusions

A series of *N*-substituted saccharins namely 2-(1,1-dioxido-3-oxobenzo[d]isothiazol-2(*3H*)-yl) acetonitrile (**2**) and (alkyl 1,1-dioxido-3-oxobenzo[d]isothiazol-2(3*H*)-yl) acetate (**3a**–**g**) were synthesized from commercially available starting materials by modified literature conditions with two different approaches; the high yield and purity of product make these conditions favorable over the previously reported ones. In addition, their chemical structures were characterized by spectroscopic techniques (^1^H-NMR, ^13^C-NMR, IR, and MS). The synthesized compounds (**2** and **3a**–**g**) were biologically evaluated. Therefore, their anti-inflammatory and antioxidant, anticancer, antibacterial, and antifungal activities were measured. The results of this study can be concluded in the following points: **I**. All the tested compounds exhibited anti-inflammatory, antioxidant, anticancer activities; **II**. Ester **3f** and nitrile **2** exhibited the highest anticancer activity toward hepatic cancer and showed good to excellent anti-inflammatory and antioxidant activities; **III**. Esters **3a**, **3d**, and **3e** exhibited the highest anti-inflammatory activity; **IV**. All the tested compounds did not exhibit any antibacterial and antifungal activities; **V.** The molecular docking calculations were evaluated as selective inhibitors of the COX-1 enzyme revealed that all the tested compounds showed inhibition activity of COX-1 by occupying some of the active sites in the enzyme with hydrogen, polar, or other bonds; **VI**. Nitriles **2** and esters **3d**, **f** with Ki = 1.86, 1.40, and 1.67 Kcal/mol^−1^, respectively, revealed the highest affinity for COX-1; **VII.** Only esters **3c**, **e**, **g** formed hydrogen bonds between the oxygen atoms of sulfonamide and oxygen of Tyr55 in COX-1 (2.88, 3.34, and 3.31 Å, respectively, and generally they exhibited low inhibition activity against COX-1; **VIII**. The electronic and the molecular structures of the compounds were refined theoretically by analysis of ***HOMO*** and ***LUMO*** energy, and the other chemical properties such as ionization potentials (***I***), electron affinities (***A***)**,** chemical softnesses (***S***), chemical hardnesses (***η***), electronic chemical potentials (**µ**), and electronegativities (***χ***) were calculated and indicated that all the tested sultams have the electron donation ability. Ester **3f** showed the highest cytotoxic activity compared to the other esters, because it needs small excitation energies which influence the biological activity of the molecule.

## Data Availability

Not applicable.

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
