# Peer review of "Synthesis, Anticancer, Antioxidant, Anti-Inflammatory, Antimicrobial Activities, Molecular Docking, and DFT Studies of Sultams Derived from Saccharin"

_molecules, 2022, doi:10.3390/molecules27207104_

Round 1
Reviewer 1 Report
In this paper, the author discusses "Synthesis, Anticancer, Antioxidant, Anti-inflammatory, Antimicrobial Activities, Molecular Docking, and DFT Studies of Sultams Derived from Saccharin”, which the author should revise and resubmit.
The following clarifications are required for this manuscript:
The table 3 is unclear, how did you calculate antioxidant activity, the author did not mention any parameters of values, why did you test CCl4 for antioxidant activity?
There is no discussion of why ampicillin was selected as the standard. The author must describe the relationship of synthesised compounds and ampicillin.
Table 2 is unclear; please explain how the results were calculated.
Author, why did you carry out Cyclooxygenase molecular docking studies? (COXs)
It is necessary to provide molecular docking for highly active vs. standard molecules.
You must present cytotoxicity activity with comparisons using normal cell lines.
Author Response
| Reviewer 1 Comments | Response |
| The table 3 is unclear, how did you calculate antioxidant activity, the author did not mention any parameters of values, why did you test CCl4 for antioxidant activity? |
We thank the reviewer for his comments
|
| There is no discussion of why ampicillin was selected as the standard. The author must describe the relationship of synthesized compounds and ampicillin | We thank the reviewer for his comment.. Ampicillin is a broad spectrum antibiotic and it is usually used as a standard regardless of the nature of the tested compounds. Additionally as mentioned in the manuscript, The compounds under investigation did not show any antimicrobial Activity. |
| Table 2 is unclear; please explain how the results were calculated. | The calculation was described in the methodology section and there was a typing mistake and corrected in methodology section |
| Author, why did you carry out Cyclooxygenase molecular docking studies? (COXs) It is necessary to provide molecular docking for highly active vs. standard molecules. You must present cytotoxicity activity with comparisons using normal cell lines |
We thank the reviewer for his comment.. Compounds 2 and 3a-g were tested as anti-inflammatory drugs (Table 2) where the proinflammatory cytokines IL-6 and TNF-α were investigated. Therefore, we tested the ability of these drugs to inhibit modulators of the inflammation process. Among these, cyclooxygenase (COXs) enzymes are importantly responsible for conversion of arachidonic acid into the prostaglandin E2 (PGE2), which is the main moderator of the inflammation process. Therefore,
|
Reviewer 2 Report
The authors reported the synthesis, properties of A series of N-substituted saccharins. These compounds are found to exhibit anti-inflammatory, antioxidant, anticancer activity. It is interesting, but the following minor revision should be considered.
1. the chemdraw schemes in Figures 1-2, Schemes 1-2 are stretched with different ratio of x and y. And thus the lines, and atoms in the figure look different. It is suggested to keep the x y ratio fixed.
2. The labels in Figure 5 is not clear. And there are box lines mixed in Figure 5.
3. The electron distribution in HOMO, LUMO orbitals will provide more clearly on why these compounds has different electron donation ability.
Author Response
| Reviewer 2 Comments | Response | |
| the chemdraw schemes in Figures 1-2, Schemes 1-2 are stretched with different ratio of x and y. And thus the lines, and atoms in the figure look different. It is suggested to keep the x y ratio fixed. | We thank the reviewer for his comment, Corrected and updated in the manuscript | |
| The labels in Figure 5 is not clear. And there are box lines mixed in Figure 5 | We thank the reviewer for his comment. Corrected and updated in the manuscript | |
| The electron distribution in HOMO, LUMO orbitals will provide more clearly on why these compounds has different electron donation ability |
We thank reviewer for their suggestion on the electron distribution of each product involve in the reaction. Now we have generated the electron distribution of HOMO and LUMO of the compounds all the tested compounds. These distributions are shown and added to the updated manuscript in fig 6. We have added the discussion in the main paper as “The spatial electron distribution of HOMO, LUMO were computed and its clearly indicate the electron donation ability.” The spatial electron distribution of each compounds (2 ,3a-g) are shown in Fig.6 |
Reviewer 3 Report
MAJOR
-in the methodology there is no description of cell cultures - it is also necessary to add full
cell line name, source/origin/reference numer
- Lack of information on statistical tests, the number of repetitions of experiments - necessary to suplement
- the reference substance for the MTT test is missing (see CCl4 for antiox/antiinfl.) – please add
- max absorbance for formazan depend on pH and cel density and has two maxima (500 and 570 nm) why you measure absorbance in 630 nm?
- what was the cell density seeding for the MTT experiment? in 4.2.2.1 you write that 50 uM (????) is a mistake - please clarify and correct
- no information on the concentrations of compounds used in individual in vitro tests
--in the results there is no link (table or chart) regarding antimicrobial activity
-change the term anti-cancer activity to cytotoxicity activity in manuscript. The anti-cancer effect could be used in the case of studying several activities related to the promotion and progression of cancer, in the presented manuscript only cytotoxicity is tested by the MTT test
MINOR
-change the abstract in such a way as to interest the reader in the further content of the article
-improve the quality of chemical formulas (Fig.1 and Fig2) as they are stretched and poorly visible
-in the result and discussion section remove subheadings (chemistry, biological activities) and replace with solid text
Author Response
| Reviewer 3 comments | Response |
|
in the methodology there is no description of cell cultures - it is also necessary to add full cell line name, source/origin/reference number |
We thank the reviewer for his comment. Corrected and updated in the manuscript (in the methodology section) "The anticancer activity of 2 and 3a-g was investigated against hepatic cancer cell (Hepatocellular carcinoma (HEPG-2) cells were obtained from ATCC via Holding company for biological products and vaccines (VACSERA), Cairo, Egypt ) with MTT (3-(4,5-Dimethylthiazol-2-yl)-2,5 diphenyltetrazolium bromide) assay was completed according to method developed by Mitryet al [64]. The cell lines were seeded in a 96-well plate at a density of 1.0x104 cells/well. at 37 C for 48 h under 5% Co2. After incubation, the cells were treated with different concentrations of compounds and incubated for 24 h." |
| Lack of information on statistical tests, the number of repetitions of experiments - necessary to supplement | We thank the reviewer for his comment. All of the experiments were carried out and repeated twice. |
|
-the reference substance for the MTT test is missing (see CCl44 for antiox/antiinfl.) – please add.
max absorbance for formazan depend on pH and cel density and has two maxima (500 and 570 nm) why you measure absorbance in 630 nm?
-what was the cell density seeding for the MTT experiment? in 4.2.2.1 you write that 50 uM (????) is a mistake - please clarify and correct -no information on the concentrations of compounds used in individual in vitro tests |
We thank the reviewer for his comment. There is no need to use standards as we compare the results with normal cells (control).
We thank the reviewer for his comment. Corrected and updated in the manuscript (in the experimental section as 570 nm ).
Corrected and updated in the manuscript
It is mentioned in table 4 (50 µmol). |
| in the results there is no link (table or chart) regarding antimicrobial activity |
We thank the reviewer for his comment. Because of all the tested compounds lack of any antimicrobial activities. Some images of the results can be provided upon request. |
| change the term anti-cancer activity to cytotoxicity activity in manuscript. The anti-cancer effect could be used in the case of studying several activities related to the promotion and progression of cancer, in the presented manuscript only cytotoxicity is tested by the MTT test; | We thank the reviewer for his comment.. Corrected and updated in the manuscript |
| change the abstract in such a way as to interest the reader in the further content of the article | We thank the reviewer for his comment. , Corrected and updated in the manuscript |
| improve the quality of chemical formulas (Fig.1 and Fig2) as they are stretched and poorly visible- | Done, Corrected and updated in the manuscript |
| in the result and discussion section remove subheadings (chemistry, biological activities) and replace with solid text | Done, Corrected and updated in the manuscript |
Round 2
Reviewer 1 Report
Still some problems for Table 3. Antioxidant activity of the tested compounds 2 and 3a-g. there is no percentage of activity present in this table
There is no normal cell lines used for cytotoxicity activity
Table 2, author should be moved for calculation part below the table.
Author not give any relationship between synthesized compounds and ampicillin.
Author Response
|
Reviewer Comments |
Response |
|
Still some problems for Table 3. Antioxidant activity of the tested compounds 2 and 3a-g there is no percentage of activity present in this table |
The percentage have been added to table 3 and updated in the manuscript |
|
There is no normal cell lines used for cytotoxicity activity
|
We followed the previously reported protocol published earlier in reference (16) (I. Elghamry, M.M. Youssef, M.A. Al-Omair, H. Elsawy. (2017) Synthesis, antimicrobial, DNA cleavage and antioxidant activities of tricyclic sultams derived from saccharin. Euro J Med Chem. 139; 107-113) |
|
Table 2: author should be moved for calculation part below the table |
It is modified and rest of calculations already described in the kit protocol to reduce repetition. |
|
Author not give any relationship between synthesized compounds and ampicillin |
As we have mentioned in the first round, Ampicillin is a broad spectrum antibiotic and is usually used as standard one. Additionally, previous published literature report have used ampicillin as standard with different organic scaffold (ex. Sak M, Al-Faiyz YS, Elsawy H, Shaaban S.” Novel Organoselenium Redox Modulators with Potential Anticancer, Antimicrobial, and Antioxidant Activities. Antioxidants (Basel). 2022 Jun 23;11(7):1231. |
Reviewer 3 Report
.
Author Response
Thank you for the reviewer, There is no comments in round 2 to response
Round 3
Reviewer 1 Report
Author still did not provide an answer about normal cell lines used for cytotoxicity testing about your compound.
There is no discussion of the relationship between structure and activity between synthesized compounds and ampicillin. Please provide more information about this relationship.
Author Response
|
Reviewer comments |
Response |
|||||||||||||||||||||||||||
|
Reviewer 1 |
Response |
|||||||||||||||||||||||||||
|
There is no normal cell lines used for cytotoxicity activity
|
Thank you for your comment. We did MTT assay on normal cell lines (WI138) and the results are displayed in the following table, and updated as supplementary file (Table S1) with the manuscript.
|
|||||||||||||||||||||||||||
|
Author not give any relationship between synthesized compounds and ampicillin |
The antibacterial activity have been repeated and Sulfamethoxazole was used instead of ampicillin. This part was updated in the discussion part of the manuscript (lines201-209). As mentioned in the introduction of the manuscript, the compounds under investigation are considered as derivatives of sulfonamide, therefore, one of the sulfa drug antibiotic, namely Sulfamethoxazole (SMZ or SMX) was used as a positive control. “The results showed that all the compounds under investigation have no antibacterial activity. Some pictures of these results are available in the editor covering letter
|